The needs of cancer patients during the COVID-19 pandemic—psychosocial, ethical and spiritual aspects—systematic review

Zapała Joanna jzapala@swps.edu.pl 1
Matecka Monika 2
Zok Agnieszka 3
Baum Ewa 4
1 Department of Postgraduate Studies, SWPS University , Warsaw , Poland
2 Department of Occupational Therapy, Poznan University of Medical Sciences , Poznan , Poland
3 Division of Philosophy of Medicine and Bioethics, Poznan University of Medical Sciences , Poznan , Poland
4 Department of Social Sciences and the Humanities, Poznan University of Medical Sciences , Poznan , Poland
Adnan Mohd
Electronic publication date: 2022 Jun 29
Publication date: 2022
Volume: 10
Electronic Location ID: e13480
Received 2021 Nov 29; Accepted 2022 May 1
Copyright: ©2022 Zapała et al.
Copyright year: 2022
Copyright holder: Zapała et al.
License: This is an open access article distributed under the terms of the Creative Commons Attribution License, which permits unrestricted use, distribution, reproduction and adaptation in any medium and for any purpose provided that it is properly attributed. For attribution, the original author(s), title, publication source (PeerJ) and either DOI or URL of the article must be cited.
License URL: https://creativecommons.org/licenses/by/4.0/

Keywords: Cancer, COVID-19 pandemic, Needs of oncologic patients

Funding: The authors received no funding for this work.

==============================
The COVID-19 pandemic resulted in unprecedented changes in the functioning of the health care system, which were connected with the occurrence of new challenges for both the health care system’s employees and for the patients. The purpose of the present article is to analyze the needs of persons with oncological diseases. Taking into account the multiple aspects of the term health, psychological, social, and existential needs of the patients were analyzed. This article is directed mainly at persons who remain in a direct therapeutic relation with a patient. It is to facilitate recognizing the needs of ill people and to increase sensitivity to the issue of maintaining or improving the well-being of patients which requires paying special attention to their psychological, social, and existential needs during the period of hindered access to the health care system. This systematic review takes advantage of quantitative and qualitative methods of text analysis with phenomenological analysis factored in. The COVID-19 pandemic resulted in the appearance of new problems in the population of oncological patients or it made the existing problems more severe. As a consequence, it made it significantly more difficult to meet their needs on various levels and sometimes it even made it impossible. It seems necessary to determine and introduce strategies to ensure that patients with oncological diseases have access to psychological and spiritual support in the period of the pandemic.

Introduction

The COVID-19 pandemic caused unprecedented social change and triggered a rapid transformation of healthcare at a global level (Tsamakis et al., 2020).

In the context of the COVID-19 pandemic, it seems important to recognise the specific needs of oncology patients. The article describes them from multiple perspectives—psychological, social and existential. Equally important is the identification of challenges that have arisen in the area of cancer care as a result of the pandemic. A closer look at these issues may prove useful in the process of providing professional multidimensional support to patients with oncological diseases.

The article is addressed to those in direct therapeutic relationship with the patient, i.e., doctors, nursing staff, psychologists, psychotherapists, physiotherapists and social workers. As members of the therapeutic team, responsible for maintaining and/or improving the well-being of patients, these individuals should also recognise and respond appropriately to patients’ needs for informational, emotional and spiritual support.

The COVID-19 pandemic is associated with experiencing a sense of uncertainty and unpredictability and a reduced sense of security. A swift and rigorous government response can alleviate psychological and emotional stress, reducing entropy and ultimately lowering certitude and helplessness. Greater trust in government and its public health interventions is associated with lower levels of depressive symptoms (O’Hara, Abdul Rahim & Shi, 2020; Tee et al., 2020b).

The introduction of rigorous measures by some countries to measure and contain the spread of COVID-19, has had positive effects not only in the area of physical health, but also in the mental health of the population. The prevalence of clinically significant depressive symptoms was significantly lower in countries where governments implemented stringent policies more promptly when compared to countries that implemented stringent lockdown measures sooner (Lee et al., 2021).

Material and Methods

A literature search was conducted in the database of MEDLINE (PubMed) and Google Scholar and PsycINFO using a combination of keywords: “cancer”, “ COVID-19”, “well-being”, “infodemic”, “internal resources”, “SARS-CoV-2”, “vaccinations”, “coping skills”. The initial search identified 11,999 publications, which were then selected on the basis of their titles and abstracts. Articles published in English and including a methodological background were considered for the study. After a qualitative text analysis, articles focusing only on clinical case reports and the economic and environmental impact of the pandemic were excluded. In particular, articles focusing on the psychosocial, ethical, and spiritual contexts of the situation of cancer patients during the COVID-19 pandemic were considered in the search. These inclusion/exclusion criteria and additional reviews of the references of the selected articles yielded 160 articles that were read and analysed.

The content of databases, MEDLINE (PubMed), PsycInfo and Google Scholar, was examined to research the impact of the pandemic on the quality of life of patients treated for oncological diseases by inputting the terms “cancer”, “covid”, “well-being”, “infodemic”, “internal resources”, “COVID-19”, “vaccinations”, and “coping skills”. The search strategy included searching for terms related to the quality of life, coping, and the pandemic. A date restriction was implemented, which focused the search on articles published since 2020.

The retrieved texts were then categorized into psychological, ethical, and spiritual issues.

The next step was to perform initial qualitative content analysis. Based on the content analysis of the abstracts, the articles that were closer to answering the research questions were selected for further analysis. The authors performed content analysis by referring to the phenomenological content analysis method. The phenomenological method has been used in psychological research for years and it also applies to content analysis (Giorgi, 2009).

Phenomenological content analysis was performed in five steps.

Step 1. Pre-reading to understand the whole

Step 2. Adopting a phenomenological psychological stance

Step 3. Dividing the data into meaningful units

Step 4. Transforming colloquial speech into psychological meaning

Step 5. Returning to the whole and moving to the overall structure (Englander & Morley, 2021).

This method made it possible, often from purely medical analyses of pandemics, to extract the meaning which was important for understanding the needs of oncology patients during pandemics.

Detailed flowchart of the data review and analysis steps. Figure 1

Relations between physical and mental health

Research suggests that poor mental health, associated for example with depressive symptoms, can adversely affect the immune system and lead to the appearance of distressing somatic symptoms (Liu, Ho & Mak, 2012; Viljoen & Panzer, 2005). Tee et al. (2021) analysed the impact of COVID-19 pandemic on physical and mental health of residents of two Asian countries—Filipinos and Chinese. Physical symptoms and low self-rated health were significantly associated with poor (adverse) mental health in both countries. Respondents from the Philippines had significantly higher levels of depression, anxiety and stress than the Chinese. They also more frequently reported physical symptoms resembling those of COVID-19 infection and were more likely to indicate: recent use of medical services and lower levels of trust with regard to medical staff; recent direct and indirect contact with COVID-19 infected individuals; concerns about COVID-19 infected family members; dissatisfaction with health information; and ostracization. The physical and psychological factors observed above were related to adverse mental health.

Figure 1 Detailed flowchart of the data review and analysis steps.

Hao et al. (2020) showed that COVID-19 patients had higher levels of neuropsychiatric symptoms compared to psychiatric patients and healthy individuals. COVID-19 patients experienced difficult emotions such as shock, fear, boredom, as well as confronting concerns related to the fact of being ill, such as discrimination, cost of treatment, need for healthcare workers. Half of the COVID-19 patients revealed clinically significant symptoms of post-traumatic stress disorder (PTSD). The prevalence of PTSD symptoms in COVID-19 patients is also indicated by reports by Bo et al. (2020). Tee et al. (2020a) in turn observed that COVID-19 pandemic had a moderate or worse psychological impact for 20% of the patients with rheumatoid arthritis or systemic lupus erythematosus who experienced symptoms such as stress, anxiety and depression in relation to this primary diagnosis.

Demographic data

Malignant tumours constitute the second cause of death in Poland. They constitute a significant health problem mainly in young and middle-aged people (25–64 years of age). This phenomenon is particularly evident in the female population, where for several years cancer has been the most common cause of death before the age of 65, accounting for 31% of deaths in young and 48% of deaths in middle-aged women (Wojciechowska et al., 2020).

The natural risk group for infections are patients with reduced resistance to infections as a result of disease (including cancer) and/or treatment (Styczyński et al., 2020). Research indicates that their risk of developing SARS-CoV-2 is twice as high as in the general population (Al-Shamsi et al., 2020). Moreover, in the population of patients with haematological cancer (non-Hodgkin’s lymphoma, chronic lymphocytic leukaemia, acute myeloid leukaemia, acute lymphoblastic leukaemia or multiple myeloma), lung cancer or metastatic cancer (stage IV), there is the highest incidence of serious cases (Dai et al., 2020). The increased risk also applies to people undergoing oncological treatment—most during chemotherapy, but also radiotherapy or surgery (Osmola, 2020). Cancer patients infected with coronavirus die three times more often than patients in the general population, for which the mortality rate is approx. 2–3%. In cancer patients, the risk is over 10 times higher (Dai et al., 2020).

As of April 26, 2021, the outbreak has been confirmed in more than 210 countries and territories. The virus has infected more than 145 million people worldwide, and the number of deaths has reached 3.1 million. The countries most affected are the United States, Brazil and India (Statista, 2021).

The report of the Alivia Oncology Foundation presents the state of Polish oncology in 2020 and diagnoses that the state of the epidemic has focused the problems observed in oncology for years (Alivia Fundacja Onkologiczna, 2020). The report mentions, among others, the lack of coordination of diagnostics and treatment, staff shortages, lack of organization of treatment in facilities, as well as patients waiting for hours in crowded corridors, failure to respect their rights, limited access to services and test results, chaos and deficits in the area of communication. Yet efforts to ensure treatment follow-up and care for cancer patients are insufficient. In the face of the pandemic, the needs of an oncological patient were no longer perceived as important (Bodnar, 2021). The authors of the report point to the lack of consistency between the various guidelines and recommendations, as well as the channels of their transmission by the public administration (Pfefferbaum & North, 2020).

A review of global research identifies stressors that undoubtedly contribute to widespread emotional anxiety and an increased risk of COVID-19-related mental disorders. These are: unclear forecasts and incomprehensible present, serious shortages of financial resources for research and treatment and protection, the need to adhere to new practices, loss of faith in health infrastructure, infodemic, as well as major and growing financial losses and conflicting messages from state authorities and international organizations (Pfefferbaum & North, 2020; Arora et al., 2020).

The National Cancer Registry reports that incidences of malignant neoplasms occur with varying frequency during the human life cycle (Krajowy Rejestr Nowotworów, 2021). The majority of malignant tumours (70% in men and 60% in women) occur after the age of 60. The risk of cancer increases with age, and is highest after the age of 80. Depending on the age group, the rate of cancer incidence varies. Among the youngest (under 20 years of age) the incidence increases slightly at a similar rate in men and women.

Among young adults (aged 20–44 years), the incidence in women is almost twice as high as in men (Tashkandi et al., 2021). According to data from Global Cancer Statistics, in 2020 there diagnosed 10.1 million new cancer cases in men (5.5 million died) and 9.2 million in women (4.4 million died) (Sung et al., 2021).

To date, many risk factors for severe disease course and mortality from COVID-19 have been identified, including old age, male sex, smoking and obesity (Tashkandi et al., 2021; Białas, Kumor-Kisielewska & Górski, 2020). The risk of death from COVID-19 in men is approximately twice as high (Bhopal & Bhopal, 2020). The risk of death from COVID-19 was highest in the elderly—this is supported by studies in the USA, Italy, Spain and China, e.g., the proportion of deaths over 75 years of age in the USA was 48.7% (Worldometer, 2021). However, the British and the Brazilian COVID-19 mutations and the arrival of the second and third waves of the pandemic influence these figures. In subsequent pandemic episodes, the virus is much more easily transmitted and infects younger people. In Brazil, COVID-19 cases among people aged 30, 40 and 50 have increased by 565%, 626% and 525% respectively since the beginning of January 2021 (Taylor, 2020). Despite this information, the people most at risk of high mortality are still the elderly and those with comorbidities—diabetes, hypertension, and cancer. The coexistence of cancer and COVID-19 is dangerous for patients. The overall 30-day mortality rate among 624 patients from five studies involving a mix of inpatient and outpatient populations was 15%. The outcome was influenced by cancer subtype (haematological versus solid), older age, male sex and recent active anticancer therapy (Desai et al., 2020).

Age is a common factor for both groups of patients—those with cancer and those at higher risk of COVID-19 infection.

Ageism is defined as stereotyping, prejudice and discrimination against people based on their age. Although ageism can be both positive and negative, in the context of the COVID-19 epidemic, a sharp increase in negative manifestations of ageism has been observed (Ayalon & Tesch-Römer, 2018). Older people, as those who already experience social isolation and loneliness, are more susceptible to the adverse effects of social distancing aimed at slowing the spread of COVID-19 (Roy et al., 2020). In an emergency situation (epidemic), there is a need for information to be transmitted quickly; older people may face significant barriers in accessing information through new media—mainly due to the “digital divide” associated with the lack of access to mobile phones or computers that allow, for example, telemedicine service. Care carried out over the Internet makes it possible for patients who are in various geographical locations to receive the same therapy and benefits from the same therapeutic interventions (Zhang & Ho, 2017).

The meta analysis carried out by Soh et al. (2020) is a source of significant support for the efficacy of Digital Cognitive Behavioral Therapy (dCBT-I) done over the Internet in the course of insomnia treatment.

Taking into consideration the needs of the general population during the COVID-19 pandemic, it is worth considering the inception of psychoeducation through the Internet and by employing smartphones in order to promote mental health and psychological interventions such as cognitive behavioural therapy (CBT) and mindfulness-based cognitive therapy (MBCT) (Ho, Chee & Ho, 2020).

Older people living in care homes are at greater risk of infection, as these institutions can act as an incubator for infection. The higher the age, the higher the risk which in turn may lead to a situation where the elderly are completely dependent on outside help—providing medication and food. The same is true if an older person is in quarantine and unable to use communication devices such as a telephone. Older people with disabilities (e.g., visual, hearing) are in a particularly difficult situation. Social distance, the use of protective masks and gloves implies a reduced ability to use previously available signals (e.g., deaf people often focus their eyes on the mouth of a person speaking). The use of protective gloves can impede the daily functioning of people with visual impairments who communicate with the environment by touch (Petretto & Pili, 2020). Isolation and staying at home can also exacerbate psychological problems in older people and affect their well-being. In this context, it is important to provide psychological support to help older people cope with a new situation such as a pandemic.

Needs and problems of cancer patients during a pandemic and after crisis and diseaster

The International Health Organization (WHO) points to the need for attention to the mental health of people with cancer, especially during a pandemic.

According to the International Agency for Research on Cancer, there will be approximately 19.3 million new cases of cancer and almost 10 million cancer-related deaths worldwide in 2020 (Sung et al., 2021).

However, studies on mental health problems in cancer patients, or patients with other chronic diseases, are scarce, which is considered a situation that needs to change (Guo et al., 2020).

The COVID-19 pandemic can be seen as a phenomenon with the features of a catastrophe, because—according to the definition of a catastrophe—it threatens personal safety, disrupts the functioning of the structure of the community and family, and causes personal and social losses, confronting the individual with requirements exceeding its resources (Ursano, Fullerton & Benedek, 2012). These features are common to the COVID-19 pandemic and the previously analyzed disasters (Gersons et al., 2020).

Pandemic, like other calamities, leads to a loss of safety and dependence on other people (related e.g., to pro-or anti-health behaviours) and is associated with experiencing chaos (Depoux et al., 2020).

It is also worth noting that the COVID-19 pandemic is a continuous disaster and responses to this disaster may vary among different populations as the disaster unfolds (Dara et al., 2005).

One study found that nearly all cancer patients (98.1%) expressed a need for information about the disease, treatment, and prognosis (Zebrack, 2009).

Chen et al. (2021) emphasizes the importance of death education, reducing pain and dying with dignity in hospice care.

Chen et al. (2021) developed and adapted the Psychological Needs of Cancer Patients Scale (PNCPS), including 23-item scale with six factors: value and esteem (five items, ie, reconsider the meaning and purpose of life), independence and control (six items, ie, private space), mental car (three items, ie, vent negative emotions), disease care (three items, ie, acquire knowledge about disease), belonging and companionship (three items, ie, spend more time at home), and security (three items, ie, living conditions be better).

It is worth noting that during the COVID-19 pandemic, the ability to meet the multidimensional needs of patients with oncological diseases was significantly reduced, which could have contributed to the deterioration of their physical, psychosocial and spiritual well-being.

Hobfoll et al. (2021) identified five basic principles of psychosocial care in relation to people who experienced catastrophe, tragedy and loss.

Gerson et al. proposed a Psychosocial Response Model that would be the basis for interventions in dealing with the challenges related to the COVID-19 pandemic. According to the Model, in response to the fear and threat experienced by patients, actions taken by professionals should be directed at strengthening the Sense of safety (e.g., by providing reliable information on the pandemic), Calming (e.g., via active communication and compassion of authorities towards patients), Self- and collective efficacy (expressed, among others, by stimulating in everyone the sense that one belongs to a group; promotion of solidarity and community activities), Connectedness (including .active implementation of digital services in education, public institutions, and other services to ensure social functioning of different groups); Hope (including through providing perspective and mitigating feelings of powerlessness and discouragement; communication about progress of treatment and vaccine developments; symbolic rituals and events to promote resilience) (Gersons et al., 2020).

The needs of cancer patients can be defined as a requirement for some action or resource that is necessary, desirable or useful to achieve optimal well-being in this patient group (Sanson-Fisher et al., 2000).

Needs assessment is therefore a key phenomenon. Firstly, it enables resources to be identified which will help the patient to function better in the situation of illness. Secondly, it allows the scale of assistance needed to be determined and prioritises the area of need, so that resources can be allocated to where the need is most urgent. Third, it enables the identification of individuals and/or subgroups of patients with higher-order needs, potentially preventing problems from occurring or reducing their severity through appropriate early intervention (Bonevski et al., 2000).

The oncology patients’ needs questionnaire considers five aspects of functioning in the disease. The first is psychological needs—related, among other things, to the expectation of emotional support and assistance of a counselling nature. The second is health information needs—including diagnosis, as well as psychological, family and financial aspects. The third aspect relates to physical needs (e.g., coping with physical symptoms and side effects of treatment) and daily life (performing daily activities and continuing treatment independently at home). The fourth is patient care and support—the support needs of cancer patients in their interactions with family, friends and healthcare professionals. The fifth is interpersonal communication—interpersonal relationships and interactions and communication styles with medical staff (Cossich, Schofield & McLachlan, 2004).

The 2018 (pre-pandemic) survey confirms that many needs of oncology patients—despite advances in the research and communication tools and medical procedures used—are still unmet (Wang et al., 2018). The deficits felt by patients were mainly in areas such as emotional support and information support (e.g., receiving information on adverse effects of treatment). Unmet needs of patients with advanced cancer were significantly associated with physical symptoms, anxiety and quality of life. Although the areas of need for cancer patients are not changing, the intensity of need depends on many factors—gender, age, socio-economic situation, coping styles, as well as the type and location of the cancer, its stage and treatment (Wang et al., 2018).

The COVID-19 pandemic has exacerbated the problems caused by the failure to meet the needs of cancer patients.

Mental health problems—depression, anxiety and coronaphobia

Confronting a stressor such as the COVID-19 pandemic outbreak is associated with a reduction in psychological resilience on an unprecedented scale and with consequences that are difficult to assess (Heitzman, 2020). According to psychological research on the effects of disasters, an increased incidence of post-traumatic stress disorder (PTSD) and anxiety and depression syndromes is expected (Di Giuseppe, Gemignani & Conversano, 2020).

The Social Insurance Institution’s (ZUS) report for 2020 shows that the number of medical certificates for self-reported illness from the category “Mental and behavioural disorders” has increased significantly. Compared to 2019, the number of certificates issued increased by 25.3% and the number of days of sickness absence increased by 36.9%. This is the fifth most frequent reason for absenteeism at work in Poland in 2020. Certificates issued for depression (that is, “Depressive episode” and “Recurrent depressive disorder”) accounted for a large share of medical certificates—385.8 thousand were issued in 2020 for a total of 7803.8 thousand days. Compared to 2019, there was an increase of 21.3% in the number of certificates issued for depression and 30.4% in the number of days of absence. The number of medical certificates for depression accounted for 26.5% of certificates issued for mental and behavioural disorders and 1.9% of all certificates issued for self-reported illness in 2020. Nearly half (44.7%) of medical certificates for depression were for people aged 35–49 years (Department of Statistics and Actuarial Forecasts of Poland, 2021). The only variable that could have caused so many people to be mentally unstable to such an extent is a pandemic.

Fear associated with the COVID-19 pandemic may not only manifest as fear of disease and death, but is also associated with socio-occupational and quarantine stress, resulting in anxiety disorders, obsessive behaviours, paranoid delusions, depression and post-traumatic stress disorder (PTSD) (Arora et al., 2020). The increased prevalence of these disorders remains related to the unreliable information about coronavirus spread through social media (Dubey et al., 2020). Specific anxiety symptoms related to the current pandemic situation have formed the picture of a new disease syndrome termed coronaphobia (Arora et al., 2020) or COVID stress syndrome (Asmundson & Taylor, 2020). The syndrome consists of five interrelated components: fear of COVID-19 infection and fear of contact with coronavirus-contaminated objects or surfaces; fear of the socio-economic consequences of the pandemic; fear of foreigners for fear of being infected; compulsive monitoring of pandemic-related information; pandemic-related post-traumatic stress symptoms (Gebbia et al., 2020). It is thought that the distant psychological effects of pandemic-related trauma, which require treatment, may affect 20% or even a larger proportion of the population (Heitzman, 2020).

The anxiety and fear resulting from the pandemic situation are compounded by the stress of cancer. A study published in Translational Psychiatry found that in a group of 6213 cancer patients during the pandemic, 23.4% were diagnosed with depression, 17.7% with anxiety, 9.3% with PTSD and 13.5% of patients experienced hostility. Among the main risk factors for mental health problems in oncology patients were: previous episodes of psychological distress; excessive alcohol consumption; more frequent worry, related to cancer treatment in the COVID-19 epidemic; feeling overwhelming psychological pressure; and higher levels of fatigue and pain. Of note, only 1.6% of patients sought psychological support (Wang et al., 2020). These data are consistent with Chinese reports from Wuhan: in a group of 1242 residents, 27.5% felt anxious, 29.3% depressed, 30% experienced sleep disturbances and 29.8% showed a passive response to COVID-19 (Fu et al., 2020).

In the run-up to the COVID-19 pandemic, levels of psychiatric symptoms in Europe were generally low. The exception was young Spanish residents with chronic diseases, who reported more symptoms than the rest of the population (Ozamiz-Etxebarria et al, 2020). During the COVID-19 pandemic period, mean stress, anxiety and depression scores in all countries were higher than normative data except Vietnam (Wang et al., 2021).

The study by Wang et al. (2021) aimed to assess the mental state of 4612 participants from 8 countries (866 from China, 982 from Poland, 619 from the Philippines, 651 from Spain, 571 from the US, 391 from Iran, 419 from Pakistan, and 113 from Vietnam) who took part in the Global Mental Health Survey during the COVID-19 pandemic. As found, Pakistan, Poland and Spain were the three countries with highest level of stress. The highest levels of anxiety were found in Pakistan, Iran and Poland. The highest levels of depression, on the other hand, were recorded among study participants from Pakistan, Poland and Spain.

The lowest scores in these areas were obtained by the inhabitants of Vietnam (Wang et al., 2021).

Health information needs

In Poland during the epidemic period, 80% of medical consultations took place via telemedicine services. According to the Government Report “Survey of Satisfaction of Patients Using Telemedicine Services with their Primary Care Physician during the COVID-19 Epidemic Period”, almost 92% of the respondents felt that obtaining a telemedicine service helped solve their health problem (National Health Fund of Poland, 2020). 43.2% expressed the belief that telemedicine service/ video advice should be one of the main channels of contact with the primary care physician (PCP) and it should be the physician who decides whether it is necessary for the patient to visit the health facility. A total of 30.4% of respondents thought that tele-counselling was appropriate when consulting chronic, previously known health problems and continuing treatment. A total of 36.3% of respondents rated the quality of an in-patient visit higher than a teleportation; according to this group, direct contact with the doctor and the opportunity to ask questions about treatment recommendations are important. A total of 6% of respondents reported that during the telemedicine service they had to seek more detailed information about the treatment.

The use of patient communication tools such as What’s App is viewed positively by doctors. What’s App appears to be a more effective tool for providing education about smoking and oral cancer compared to conventional audio-visual aids (Nayak et al., 2018). Gebbia et al. (2020) found that approximately 82% of physicians expressed appreciation for such a solution; the mean satisfaction score for its use was 7.8 on a 0–10 scale. A study by Hasson et al. (2021) showed patients perceive that telemedicine as safe and effective, not compromising medical care or the patient-doctor relationship. The inclusion of telemedicine services was found to be beneficial for patients under medical control after active cancer treatment. Most respondents felt that their concerns had been addressed and their needs met. Moreover, the patients appreciated the eye contact with the doctor, they felt that they were listened to, and the feedback—the doctor’s explanations and treatment plan—was provided in a clear manner. Patients felt safe and declared that their privacy was respected.

In a study by Tashkandi et al. (2021) on new ways of communicating with oncology patients, it was found that a telephone call was the preferred method of communication (92% of indications), followed by an electronic patient portal (75% of indications), a mobile app (76% of indications), telemedicine (73% of indications) and text messaging (72% of indications). The majority of patients (97%) preferred the use of an electronic patient health data system during appointments, 95% of patients preferred it for delivering medications and viewing laboratory tests, and 92% of patients preferred it for requesting medical certificates.

Although telemedicine has gained acceptance by both the majority of patients (Millar et al., 2020) and medical staff, its potential risks are also noted. Moreover, telephone communication about a patient undergoing isolation in unstable or severe clinical conditions is a difficult task for doctors, nurses and family members due to the emotional burden.

High levels of COVID-19 symptoms and the overlap between COVID-19 and cancer-related symptoms are a challenge for clinicians who assess and select oncology patients for COVID-19 testing. For patients undergoing active treatment, clinicians face challenges in assessing and managing symptoms that, prior to COVID-19, would have been attributed to acute toxicity resulting from cancer treatment or specific to cancer survivors (Miaskowski et al., 2021). The lack of direct contact between patient and oncologist can adversely affect patient well-being, motivation and sense of security, especially during periods of uncertainty such as a pandemic (Elkaddoum et al., 2020).

Telemedicine services have been an important part of medical care during the pandemic, not least because of the problem of moving around, getting to health facilities. Transport was another significant practical problem identified. Concerns about the use of public transport, the cancellation of many transport services due to COVID-19, as well as transport disruptions prevented patients from accessing oncology treatment and medical appointments (Edge et al., 2021).

In the study by Tashkandi et al. (2021) cited above, it was found that 7% of subjects were unable to attend treatment due to transport problems. Those participating in the Alivia Foundation study also reported the problem of transport to medical units. Therefore, in response to the significant increase in infections and the second wave of the epidemic, at the beginning of December 2020, the Alivia Foundation resumed the #ONKOTAXI action—free transport by taxi, observing safety rules, to treatment centres for oncological patients (Alivia Fundacja Onkologiczna, 2020).

Patient care and expected support

UK researchers from the CovidSurg Collaborative have attempted to estimate the scale of operations that have not taken place due to coronavirus. As they found, for 12 weeks after the COVID-19 outbreak in Europe, 72.3% of all scheduled operations were cancelled. Oncological procedures were the second most numerous group among them: 37.7% of them were postponed (i.e., 2,324,070 out of 6,162,311). In Poland, 22,656 surgeries were cancelled during the period indicated (COVIDSurg Collaborative, 2020).

Limited access to doctors, cancelled operations and examinations may be due not only to the limitations of the health system, but also to patients’ decisions.

Patients with cancer are a group at increased risk of contracting COVID-19. Their weakened immunity is due to factors such as tumour burden and active oncological treatment (cytotoxic chemotherapy, radiotherapy), undergoing transplantation, and using immunosuppressive drugs (Abdihamid et al., 2020). In this context, the fear of cancer patients, their fear of being infected with COVID-19, is understandable, as evidenced by the study of Vanni et al. (2020) As found, both breast cancer patients and those with suspected lesions were more likely to abandon treatment and surgery during the pandemic period. The risk of infection was the main reason for abandonment.

In a study on the use of telemedicine services among oncology patients, it was found that the most common questions asked by patients were about rescheduling a test (37%) and the possibility of continuing anticancer therapy (29%) (National Health Fund of Poland, 2020). A statistically significant association was observed between questions about study rescheduling and study/treatment cancellation and patient age over 75 years. Questions about patients’ well-being indicated that they felt primarily fear, anger and sadness (57%).

In Poland, since the announcement of the pandemic and lockdown, the number of issued DiLO Diagnostic and Oncological Treatment cards has decreased. Data from the National Health Fund (NFZ) show that already in the first month of the epidemic in Poland (March 2020) there were 1780 fewer of them than a year earlier. In April—when restrictions related to the COVID-19 pandemic were already in force in Poland—38% (8377) fewer cards were issued than in April 2019, and in May 2019 30% fewer than a year earlier (Narodowy Fundusz Zdrowia, 2020). There is a high likelihood that patients who have not been screened for various reasons will end up in hospital with advanced, disseminated cancer after the pandemic.

An increase in anxiety related to cancer care is also observed in their caregivers, especially in parents of children with cancer. 69.6% of parents/carers did not perceive the hospital as a safe place (Darlington et al., 2020).

Human relations—quarantine, isolation in hospital

A Chinese study of people in isolation indicates that 34.8% of patients with oncological diseases experienced increased anxiety or depressive symptoms; the severity of these conditions was higher among them compared with participants without chronic diseases, also in quarantine (Guo et al., 2020). Prolonged isolation has a negative impact on psychological response, promoting post-traumatic stress symptoms, feelings of confusion and anxiety (Maugeri et al., 2020). Patients with COVID-19 treated in closed wards, intensive care units or other isolated settings are deprived of opportunities for direct contact with loved ones. Other sources of companionship and support (volunteers, clergy) have largely been reduced or made unavailable because of the pandemic. Medical staff face and assume the burden of providing psychological support to patients (Münch et al., 2020). The psychological needs associated with hospital isolation revolve primarily around a sense of social solidarity with loved ones and others; obtaining advice on individual therapy; support in coping with the experience of reduced living conditions and confronting one’s own mortality; support in experiencing grief over the losses experienced; support in coping with anxiety about loved ones; and coping with fear, stress and depression (Makowska & Poprawa, 2001). The perspective of alienation of both patients and medical staff was included in the Italian study. Indeed, isolation is embedded both in the experience of patients separated from their loved ones and in the experience of those providing treatment and care to patients with COVID-19 (Gruppo di Lavoro Intersocietario “ComuniCoViD”, 2020).

Internal sources of support in cancer and the COVID-19 pandemic

Morhaf Al Achkar, a doctor affected by cancer, compared being a cancer patient in the age of the pandemic to being imprisoned (Al Achkar, 2020). He also noted that despite the burdens experienced by cancer patients, they are perhaps better equipped than others to deal with existential threat. By confronting their own mortality through the experience of cancer illness, these individuals are aware of the fact that life is finite, and this helps them to develop the ability to live in the ’here and now’.

Oncology patients in the process of coping with the disease refer to response styles depending on their temperament, personality (Makowska & Poprawa, 2001). Effective coping is associated with a decrease in experienced stress (improvement of well-being), reduction of anxiety and depression and an increase in hope. Basińska analysed coping styles of women with advanced cancer of the reproductive organs. As she found, higher coping efficacy is associated with the use of positive illusions, especially such as unrealistic optimism and excessive perception of control. These mechanisms promote effective adaptation to stressful events, including in extreme situations. Better adaptation to illness has also been observed with limited use of avoidance and activity strategies in treatment decision-making.

Emotional de-stressing strategies were in turn associated with poor adaptation to cancer. Palliative patients were found to avoid threatening information more often than patients undergoing radical management, which was positively defensive. Among older patients diagnosed with cancer (lung, breast, bowel cancer), positive mood and higher levels of hope were related to internalised religiosity and spirituality. A sign of a coping attitude was an orientation towards life and recovery. Hope and acceptance of the disease as a situation that cannot be controlled or changed influenced the undertaking of treatment-oriented activities (Basińska, 2000).

Coping styles are relatively stable and describe how individuals habitually interact with their environment. It can therefore be assumed that cancer patients who used familiar coping styles in cancer will also refer to them in relation to the pandemic. In a study of the coping strategies of coronavirus cancer survivors, coping strategies of acceptance (96.7%), distraction (93.3%) and engagement (93.3%) predominated (Galica et al., 2021). In a study of ways to reduce stress and anxiety and enhance well-being among pandemic cancer patients, it was found that they mainly refer to sources of support such as interpersonal relationships, spirituality, mindfulness techniques and contact with nature.

Family and friends as informal sources of support

Although isolation helps to achieve the goal of reducing infections, limited access to family, friends and other social support systems is associated with feelings of loneliness, exacerbating psychological problems such as anxiety and depression (Hiremath et al., 2020). Being rooted in a social network and the associated ability to sustain face-to-face interactions with people allows the individual to receive support before the need for it is actualised. COVID-19 has significantly reduced, and sometimes—prevented—this natural, intuitive process of mutual support.

Two studies are of interest—an English study and a Dutch study—in which cancer patients declared that the pandemic had not negatively affected their well-being and was not associated with an increased sense of unmet needs. In the UK study, overall levels of need did not increase, but individual needs did change. Cancer patients reported significant reductions in anxiety and improvements in quality of life after the pandemic outbreak, while those supporting patients indicated reduced quality of life and increased anxiety, stress and depression. Initially, the pandemic did not adversely affect the psychological well-being of cancer patients. The need to update intra-family resources in response to the lost availability of institutional support may explain the significant and detrimental impact on well-being and quality of life of informal caregivers of cancer patients (Hulbert-Williams et al., 2021). Similar conclusions were drawn from a Dutch study that assessed well-being in families with children with cancer (Van Gorp et al., 2021). In a group of almost 800 people, no differences were observed between the periods before COVID-19 and the early period of the COVID-19 pandemic regarding health-related quality of life and children’s fatigue levels. The additional stress of COVID-19 did not impair the psychosocial functioning of children with cancer and their caregivers. It can be speculated that the reported lack of negative changes in children’s and caregivers’ functioning remained, among other things, related to parents’ experience in coping with traumatic stress and adequate care and support. Both studies emphasise the active, supportive role played by the cancer patient’s immediate support environment—family, friends. These findings are consistent with reports from studies of women with breast cancer (Brivio et al., 2021).

Spirituality and inner resources

Research shows that religiousness and spirituality are broadly resorted to in critical moments in life. Growing interest in spirituality and religion finds reflection in conceptualising spirituality and religion as separate, yet overlapping constructs (Hodge, 2003). A Brazilian study determined that religion and spirituality play a significant role in alleviating suffering, influencing health parameters and minimising the consequences of social isolation. In this context, it is worthwhile to underline the significance of securing financial resources for maintaining continuity of religious and spiritual support during the pandemic, as well as of training the medical personnel to solve patients’ problems in this field (Lucchetti et al., 2021; Del Castillo, 2021). Owing to pandemic-related difficulties in maintaining direct social contacts, spiritual support for patients, their families, and medical personnel has been drastically limited, both qualitatively and quantitatively. It was provided by means of various technological devices. No mention of developing institutional strategies aimed at providing spiritual support has been found (Papadopoulos et al., 2021). For this reason, a number of initiatives offering remote spiritual support have evolved, such as ChurchInAction in the Philippines (Del Castillo, Biana & Joaquin, 2020), or ones that prompt medics to support their patients also in the spiritual sphere (Roman, Mthembu & Hoosen, 2020).

Mindfulness

Practising mindfulness can help to lower the level of anxiety. It is also conducive to mitigating the effects of traumatic experiences and catastrophes, by lessening the negative affect, as well as syndromes of depression and post-traumatic stress disorder (PTSD) (Boyd, Lanius & McKinnon, 2018).

Potentially, it can also prevent psychological stress associated with COVID-19 and quarantine-related encumbrances and losses (Fu et al., 2020). Polish studies on the influence of mindfulness upon anxiety disorders associated with the pandemic, which involved 301 persons meeting the adjustment disorder criteria, determined that application of mindfulness-based interventions significantly diminished the intensity of sickness syndromes. Increased detachment from own syndromes—negative thoughts, emotions and sensations (cognitive defusion) has also been observed in participants of the study. What is important, the increased ability to detach oneself from one’s own difficult experiences turned out to be a change mechanism leading to a decrease in the level of anxiety and depression syndromes (Holas et al., 2021).

Contact with nature

In the Japanese study, an attempt was made to establish the relation between five mental health-related areas (depression, satisfaction with life, subjective feeling of happiness, self-esteem, and loneliness) and such aspects, as the frequency of making use of green areas, as well as the sight of greenery from one’s windows. It was determined that the frequency of using green areas and opportunity to view greenery from one’s window were associated with increased self-esteem, satisfaction with life and subjective feeling of happiness, as well as with diminished level of depression, anxiety and loneliness. The Japanese findings indicate that regular contact with nature may contribute to lessening several different problems in the area of mental health. Due to the observed increase in the number of mental health disorders and the negative influence of COVID-19 pandemic on public mental health, these findings carry vital political message, as they prove that green enclaves in towns can be perceived as “natural solutions” to problems of public health (Soga et al., 2021),

Forest baths reduce biomarkers associated with cardiovascular disease, reduce blood pressure, urinary adrenaline and serum and salivary cortisol levels. Also, Horticultural Therapy (HT) improves inflammatory markers (IL-6) supports neuroprotection and prevents dementia. Older adults with depression, rheumatoid arthritis and cancer also benefit from HT (Ng et al., 2018; Ng et al., 2021).

Ecosystem services, in particular exposure to the natural world (blue–green spaces), exert positive influence upon mental health and well-being. An international survey (5,218 responses from nine countries) determined that while lockdown exerted profound influence on mental health, contact with nature helped people to cope with the negative effects of seclusion. In Spain, persons under strict lockdown (3,403 responses) noticed that contact with nature helped them to cope with their isolation; an opportunity to make use of open space, as well as blue–green view from their windows also contributed to experiencing more positive emotions (Pouso et al., 2021).

According to findings, access to bodies of water or blue spaces can influence one’s health and well-being in a positive way (McDougall et al., 2020). The feeling of lack of access to green urban space was alleviated by a more open view of the natural environment, and—partly—a green view outside the window (Ugolini et al., 2021). It turned out that during the pandemic, when the majority of time was spent inside the house, time spent out in the open brought about a clear increase of positive affect and decrease of negative emotions. Exercise, walks, gardening, hobbies and childcare are activities that exert positive influence upon social well-being (Lades et al., 2020).

Vaccinations

Older people tend to respond less well to vaccination due to the ageing immune system. According to Our World in Data, as of 29 April 2021, 254 million people worldwide have been vaccinated with a full dose—representing 3.3% of the human population. Most citizens have been vaccinated in Israel—over 56% of the population, the United Arab Emirates—39.3%, Chile—34.1% and the USA—almost 30%. In Poland, 7.2% of the population has been vaccinated with a full dose (Our World in Data, 2021). In Poland, as of 22 January 2021, vaccination of people 70+ (in Poland this is about 4.5 million people) began, and as of 15 March 2021, vaccination began, among others, of patients with cancer who were treated with chemotherapy or radiotherapy after 31 December 2019, and those who were diagnosed with cancer but did not start treatment (Serwis Rzeczpospolitej Polskiej, 2021). Information about the side effects of vaccines, difficulties in contacting the registration system (difficulty in connecting to the telephone line of vaccination coordinators, distant dates, distant vaccination sites), slowdowns in the supply of vaccines, as well as information about post-vaccination deaths or the origin of vaccines may have increased anxiety, reluctance to vaccinate those who should be vaccinated most quickly.

Patients’ fear of vaccination stems from the problems outlined above. Limited contact with the health care system, rapid teleportation is not always sufficient to clarify patient concerns. The situation of oncology patients during the pandemic is particularly difficult. Systemic support, unambiguous and reliable information, access to vaccines and the opportunity to start and continue treatment are not all cancer patients need. A holistic view of their wellbeing is also crucial, and teaching patients and their relatives to use their own internal resources can help with this. It is also important to learn to use alternative ways of keeping in touch with loved ones - online meetings or phone calls.

COVID-19 is likely to be one of the diseases that will be with people for a long time to come, so it is important to learn to live in a world where the threat of this virus becomes part of everyday experience.

Conclusions

Cancer patients, in addition to the risks associated with the disease itself and the associated treatment, are confronted with many psychological, social and existential challenges. The diagnosis of cancer itself is a borderline experience. Undergoing therapy, which is a long-term process, is connected with the necessity to introduce changes in all areas of functioning—cognitive (connected, among others, with the perception of time, sense of individual life), emotional (coping with strong difficult emotions), social (exposure to stigma, sense of loneliness, etc.), spiritual (connected with questions and reflections of eschatological nature). In this context, patients need multidimensional professional support, aimed both at restoring or maintaining somatic health potential and focused on strengthening psychological resources.

The pandemic has significantly complicated and hindered the process of identifying and adequately responding to the needs of patients with oncological diseases. Consequently, it has exacerbated the suffering of patients and their loved ones. The already extremely difficult experience of coping with the disease and treatment has been accompanied by anxiety resulting from the lack of availability of help, not only to improve or maintain health in the long term, but above all to provide immediate, life-saving help.

Further directions of research should include searches geared towards ensuring that patients with oncological diseases have the option of constant access to psychological help realized in different ways—crisis intervention, psychoeducation, psychological rehabilitation, and psychotherapy. The possibility of obtaining emotional, informational, and spiritual support is significant in the whole life cycle and it is indispensible in extraordinary situations such as crises, pandemics, or catastrophes.

Additional Information and Declarations

Competing Interests

Author Contributions

Data Availability

The authors declare there are no competing interests.

Joanna Zapała analyzed the data, authored or reviewed drafts of the article, and approved the final draft.

Monika Matecka analyzed the data, authored or reviewed drafts of the article, and approved the final draft.

Agnieszka Zok analyzed the data, authored or reviewed drafts of the article, and approved the final draft.

Ewa Baum analyzed the data, authored or reviewed drafts of the article, and approved the final draft.

The following information was supplied regarding data availability:

This article is a literature review.

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
