# Peer review of "The needs of cancer patients during the COVID-19 pandemic—psychosocial, ethical and spiritual aspects—systematic review"

_PeerJ, doi:10.7717/peerj.13480_

## Round 0.1 · original submission · Major Revisions

Despite addressing a topic of utmost importance in these times, manuscript needs substantial revision and additional work in order to appreciate the quality for publication. Manuscript has been thoroughly reviewed by the experts and have commented against the acceptance of manuscript in its current form due to serious methodological and conceptual concerns. Please see the detailed comments made by the reviewers.

Reviewers have also suggested and recommended discussing a few findings, please make sure that they are directly relevant to the work and will contribute to the manuscript effectively.

Reviewer 1 has suggested that you cite specific references. You are welcome to add it/them if you believe they are relevant. However, you are not required to include these citations, and if you do not include them, this will not influence my decision.

Furthermore, I have noticed that English language must be improved and thorough editing is required. Please revise the manuscript taking help from a colleague who is proficient in English and familiar with the subject matter, who can review your manuscript, or contact a professional editing service to review your manuscript. Revise and resubmit accordingly.

Reviewer 1 ·

Basic reporting

I have the following comments for the authors to address. I am happy to review this paper again

1) The authors used "SARS-CoV-2". It is not the most updated term. The authors should use COVID-19.

2) The following statement "The article is intended for professionals who provide support to patients: healthcare personnel, therapists, NGO employees, and social workers. Accurate identification of chronic condition patients’ new needs should also be important for entities in charge of organisation of medical help and assistance on all levels". The above statements are too general and please remove.

3) The authors stated "SARS-CoV-2 pandemic has caused unprecedented social changes and started rapid healthcare transformation on the global level [1]". I recommend the authors to state the governments' response based on the following study:

Government response moderates the mental health impact of COVID-19: A systematic review and meta-analysis of depression outcomes across countries. J Affect Disord. 2021 May 27;290:364-377. doi: 10.1016/j.jad.2021.04.050. Epub ahead of print. PMID: 34052584.

4) The authors stated "Its outbreak did not lower the rates of infection with other..." This statement does not make sense and suggests to be removed.

5) The authors stated "Even though the World Health Organisation (WHO) points to the necessity to pay attention to mental health of this population," I suggest the authors to discuss the link between physical and mental health:

Impact of the COVID-19 Pandemic on Physical and Mental Health in Lower and Upper Middle-Income Asian Countries: A Comparison Between the Philippines and China. Front Psychiatry. 2021 Feb 9;11:568929. doi:
10.3389/fpsyt.2020.568929. PMID: 33633595; PMCID: PMC7901572.

6) Before talking about COVID-19 impact on cancer, please discuss how the mental health of COVID-19 patients and other chronic diseases based on the following studies:

A quantitative and qualitative study on the neuropsychiatric sequelae of acutely ill COVID-19 inpatients in isolation facilities. Transl Psychiatry. 2020 Oct 19;10(1):355. doi: 10.1038/s41398-020-01039-2. PMID: 33077738.

Psychological State and Associated Factors During the 2019 Coronavirus Disease (COVID-19) Pandemic Among Filipinos with Rheumatoid Arthritis or Systemic Lupus Erythematosus. Open Access Rheumatol. 2020 Sep 22;12:215-222. doi: 10.2147/OARRR.S269889. PMID: 33061689; PMCID:PMC7520098.

7) Under Contact of Nature, please discuss discuss the findings of the following studies:

Effects of Horticultural Therapy on Asian Older Adults: A Randomized Controlled Trial. Int J Environ Res Public Health. 2018 Aug 9;15(8):1705. doi: 10.3390/ijerph15081705. PMID: 30096932; PMCID: PMC6121514.

Social connectedness as a mediator for horticultural therapy's biological effect on community-dwelling older adults: Secondary analyses of a randomized controlled trial. Soc Sci Med. 2021 Sep;284:114191. doi: 10.1016/j.socscimed.2021.114191. Epub 2021 Jul 9. PMID: 34271401.

Experimental design

The authors should indicate in the title and the abstract that this is a systematic review.

Validity of the findings

1) The authors stated "serious obstacles in reaching information by means of the new media – mainly owing to “the digital gap” associated with lack of access to mobile phones or computers, which make use of e.g. online consultation possible". Please provide more details including internet cognitive behavior therapy by adding the following references:

The most evidence-based treatment is cognitive behaviour therapy (CBT), especially Internet CBT that can prevent the spread of infection during the pandemic.

Use of Cognitive Behavior Therapy (CBT) to treat psychiatric symptoms during COVID-19:
Mental Health Strategies to Combat the Psychological Impact of COVID-19 Beyond Paranoia and Panic. Ann Acad Med Singapore. 2020;49(3):155‐160.

Cost-effectiveness of iCBT:
Moodle: The cost effective solution for internet cognitive behavioral therapy (I-CBT) interventions. Technol Health Care. 2017;25(1):163-165. doi: 10.3233/THC-161261. PMID: 27689560.

Internet CBT can treat psychiatric symptoms such as insomnia:
Efficacy of digital cognitive behavioural therapy for insomnia: a meta-analysis of randomised controlled trials. Sleep Med. 2020 Aug 26;75:315-325. doi: 10.1016/j.sleep.2020.08.020. Epub ahead of print. PMID: 32950013.

2) The authors stated "Psychiatry showed that in a group of 6213 patients who had cancer in the period of the pandemic, 23.4% were diagnosed with depression, 17.7% – with anxiety, 9.3% – with PTSD, and 13.5% of patients felt hostility". The authors should compare the prevalence of depression, anxiety and PTSD with general population based on the following study:

A chain mediation model on COVID-19 symptoms and mental health outcomes in Americans, Asians and Europeans. Sci Rep 11, 6481 (2021). https://doi.org/10.1038/s41598-021-85943-7

Reviewer 2 ·

Basic reporting

The article has clear text and conforms to professional standards of courtesy and expression. Some places of improvement are:
Line 55, 56 – Using incl. and ca. is not very common in scientific articles. I suggest at least for the first time use, please use the full form.
Line 66, 67 – Is this a new paragraph? It is too small to be a paragraph on its own.
Line 78 – As ‘of’
Other comments are annotated in the pdf.
The article includes sufficient introduction and background to demonstrate how the work fits into the broader field of knowledge.

Experimental design

This article's content is within the aims and scope of the journal.
The authors describe the field and explain why this review is needed and explain the relevance to readers in the field, and associated areas by citing previous reviews.
However, the methodology excludes the articles and data not published in English. I feel that the authors might be losing some crucial data that is not published in English. COVID19 being a pandemic, there might be some crucial data from other countries.
Additionally, as annotated in the pdf, in lines 78, 229, 560, the authors report statistics from early 2021. In the current pandemic, I think it would be relevant and accurate to report the latest data and statistics.

Validity of the findings

no comment

Additional comments

In the context of a pandemic like COVID19, with ever-changing statistics, it is crucial to report the latest data. The authors in this article mostly report data and numbers from early 2021. Updating and re-visiting the results might be of great value to this literature review.

Annotated reviews are not available for download in order to protect the identity of reviewers who chose to remain anonymous.

Reviewer 3 ·

Basic reporting

The article covers a very important topic, the psychosocial, ethical and spiritual needs of cancer patients during the SARS-CoV-2 pandemic. I complement the authors for covering this important topic. However, I have some major concerns regarding the reporting and design of the study.
- Abstract: the abstract is missing structured information about the methodology used, a clear objective and structured information about the major results and conclusions.
- The introduction, abstract and title describe the objective of this review article slightly different. I think the objective is to assess the psychosocial, ethical and spiritual needs of cancer patients during the pandemic. Please be consequent in the formulation of the objective. For example, in the introduction the authors also mention that they are investigating the medical needs.
- My major concern is that the previous research on a) psychosocial needs after crisis, and b) psychosocial needs of cancer patients is missing in the introduction. What has previous research identified as main psychosocial needs after crisis? And what has previous research identified as important needs for cancer patients (for example Chen at al., (2021) developed an instrument measuring important psychosocial needs for cancer patients based on Maslow’s hierarchy of needs)? Regarding the psychosocial needs after crises and disasters many research on this topic exists (for instance a standard work in this area is Hobfoll et al., 2009) as well as international guideless (for example TENTS guidelines (Bisson et al., 2020) and recent WHO guidelines on psychosocial support). This body of literature can serve the authors as it can function as a framework to identify the most important needs that might be currently missing for cancer victims. When this framework based on previous research is made clear in the introduction, the specific needs for cancer victims in this specific crisis will follow logically in the result section.

Experimental design

I have some major concerns regarding the methodology.
- The method section is too brief and is missing detailed and vital information. I miss a clearly described search strategy and screening strategy. Also, there is no information about the methodological strength of the studies included. In this way, it will not be possible to replicate this study or to clearly assess the strength of the results of this study.
- It is unclear why the used keywords in the search strategy were included. Why are keywords not included that are directly related to the objective, namely psychosocial needs, spiritual needs and ethical needs (or related keywords such as social support, supportive interventions)? Keywords are now used that are not previously introduced to the reader, such as ‘coping skills’.
- Psychosocial needs and care is also a psychological topic. It is recommended to include PsycINFO as a database.
- Please provide a detailed overview of inclusion and exclusion criteria that makes it clear to the reader how the authors arrived at the inclusion of 160 articles (starting with 11999 articles).
- Please provide a section 'categorization of findings' in the method section that will describe how the results will be categorized and explains to the reader why this categorization had been chosen.

Validity of the findings

- The article as a whole is very lengthy and this makes it for the reader difficult to read. It will help if the information, especially the results and conclusion section, can be described more structured with subheadings and a clear framework/outline introduced in the introduction or method section. The information can also be described more concise. A table with key results and conclusions can also help the reader.
- Due to the lack of 1. a clear objective and b. a clearly described methodology, it is not clear to the reader where the results come from and what the strength of the findings is. For example, the section demographic data reads as an introduction instead of research findings on psychosocial needs. And why is mindfulness mentioned as important for treating PTSD (also a citation is missing here) while this not at all first choice of treatments regarding to the guidelines for treatment of PTSD. This should be discussed and a discussion is missing To me as a reader, it feels like the results and conclusion are missing a clear categorization of findings and discussion of the validity of the findings. It now reads as scattered parts of results.
- The conclusion does not identify unresolved questions / gaps / future directions.

---

## Round 0.2 · accepted · Accept

Manuscript is significantly improved by the authors and now can be accepted in its current form.

Reviewer 1 ·

Basic reporting

I am satisfied with the amendments.

Experimental design

I am satisfied with the amendments.

Validity of the findings

I am satisfied with the amendments.

Additional comments

I am satisfied with the amendments.

Reviewer 2 ·

Basic reporting

no comment

Experimental design

no comment

Validity of the findings

no comment

Additional comments

The authors have taken care of the suggested corrections.